# Quantization Error as a Metric for Dynamic Precision Scaling in Neural Net Training

**Ian Taras & Dylan Malone Stuart**
Department of Electrical and Computer Engineering
University of Toronto
Toronto, ON, Canada
`{tarasian, malones2}@ece.utoronto.ca`

## Abstract

Recent work has explored reduced numerical precision for parameters, activations, and gradients during neural network training as a way to reduce the computational cost of training (Na & Mukhopadhyay, 2016) (Courbariaux et al., 2014). We present a novel dynamic precision scaling (DPS) scheme. Using stochastic fixed-point rounding, a quantization-error based scaling scheme, and dynamic bit-widths during training, we achieve 98.8% test accuracy on the MNIST dataset using an average bit-width of just 16 bits for weights and 14 bits for activations, compared to the standard 32-bit floating point values used in deep learning frameworks.

## 1 Introduction

It is well established that neural networks, though ordinarily trained using 32-bit single precision floating point representation, can achieve desirable accuracy during inference with reduced precision weights and activations (Judd et al., 2015) (Mishra et al., 2017) (Courbariaux et al., 2015) (Hubara et al., 2016). These reduced precision networks are amenable to acceleration on custom hardware platforms which can take advantage of lower bit-widths in order to speed up computation (Na & Mukhopadhyay, 2016) (Gupta et al., 2015). Reduced precision strategies are not typically applied during back-propagation whilst training, as this can lead to heavily reduced accuracy or even non-convergence.

Recent work has shown that dynamic precision scaling, a technique in which the numerical precision used during training is varied on-the-fly as training progresses, can achieve computational speedups (on custom hardware) without hampering accuracy (Na & Mukhopadhyay, 2016) (Courbariaux et al., 2014). DPS uses feedback from the training process to decide on an appropriate number representation. For example, Na & Mukhopadhyay (2016) suggest starting with reduced precision, and increasing precision dramatically whenever training becomes numerically unstable, or when training loss stagnates.

In this paper, we present a novel DPS algorithm that uses the stochastic fixed-point rounding method suggested by Gupta et al. (2015), the dynamic bit-width representation used by Na & Mukhopadhyay (2016), and an algorithm that leverages information on the quantization error encountered during rounding as a heuristic for scaling the number of fractional bits utilized.

## 2 Fixed Point Representation and Quantization/Rounding

Fixed point numbers are represented by a fractional portion appended to an integer portion, with an implied radix point in between. We allow our fixed point representation to use arbitrary bit-width for both the integer and fractional parts, and represent the bit-width of the integer part as $IL$ and the bit-width of the fractional part as $FL$. We denote a given fixed point representation, then, as $\langle IL, FL \rangle$. DPS modifies $IL$ and $FL$ on-the-fly during training.

Inspired by Gupta et al. (2015), we use stochastic rounding during quantization of floating point values to $\langle IL, FL \rangle$, as it implements an unbiased rounding.

Our algorithm employs a dynamic bit-width, dynamic radix scheme in which $IL$ and $FL$ are free to vary independently. Note that with the alternative fixed bit-width scheme, $IL$ and $FL$ are inter-dependent as increasing one necessitates a decrease in the other.

# 3 DYNAMIC PRECISION SCALING ALGORITHM

Here we formally introduce our novel DPS algorithm which leverages average % quantization error as a metric for scaling fractional bits. Quantization error is calculated on a per-value basis as in Equation 1. Quantization error % is accumulated and averaged over all round operations – this is the metric used when scaling $FL$.

$$E_\% = \frac{|x_{out} - x_{in}|}{x_{in}} \times 100 \qquad (1)$$

Table 1 frames this work in relation to prior work in the area.

---

**Algorithm 1** Dynamic Precision Scaling with Quantization Error

---

**Input**: Current Integer Length: **IL**, Current Fractional Length: **FL**
      Overflow Rate: **R**
      Average % Quantization Error: **E**
      Maximum Overflow Rate: **R_max**
      Maximum Average Quantization Error: **E_max**
**Output**: $\langle IL, FL \rangle$ for the given attribute (Weights, Gradients, or Activations).

  1:   **Begin**
  2:      **if** R > R_max:
  3:         $IL \leftarrow IL + 1$
  4:      **else**
  5:         $IL \leftarrow IL - 1$
  6:      **if** E > E_max:
  7:         $FL \leftarrow FL + 1$
  8:      **else**
  9:         $FL \leftarrow FL - 1$
10:   **End**

---

Table 1: Summary of related work

| Authors | Fixed point format (bit width, radix) | Scaling | Rounding | Precision Granularity |
|---|---|---|---|---|
| (Na & Mukhopadhyay, 2016) | (Dynamic, Dynamic) | Convergence/ Training Based | Nearest | Per-Layer |
| (Courbariaux et al., 2014) | (Fixed, Dynamic) | Overflow Based | Nearest | Per-Layer |
| (Gupta et al., 2015) | (Fixed, Fixed) | None | Stochastic | Global |
| Essam et al. (2017) | (Fixed, Dynamic) | Overflow Based | Stochastic | Global |
| (Köster et al., 2017) | (Fixed, Dynamic) | Predictive Max-Value | N/A | Per-Tensor |
| Ours | (Dynamic, Dynamic) | Overflow and Quantization Error Based | Stochastic | Global |

# 4 EXPERIMENTS

In order to perform evaluations, we emulate a dynamic fixed point representation by using custom Caffe layers that quantize/round the native floating point values to values that are legal in our fixed point format. In our study, we consider training a neural network using stochastic gradient descent with dynamically scaled precision for weights, activations, and gradients during both the forward

(inference) and backward pass. As per Na & Mukhopadhyay (2016), we quantize weights, biases, activations, and gradients at the appropriate pass through the network, and update the precision on-the-fly during training on each iteration.

We train LeNet-5 on the MNIST dataset using Caffe and our custom rounding layers and DPS algorithm (Lecun et al., 1998). We use a batch size of 64, and train for 10,000 iterations. We use an initial learning rate of 0.01, momentum of 0.9, a weight decay factor of 0.0005, and scale the learning rate using $lr = lr_{init} * (1 + \gamma * iter)^{-pow}$, where $\gamma = 0.0001$ and pow = 0.75. We update IL and FL once each iteration, and use $E_{max} = R_{max} = 0.01\%$.

We compare our results to a baseline network trained on the same dataset with the same hyperparameters, but using full-precision floating point for all attributes. We also compare against a non-dynamic fixed point representation that uses 13 bits for weights and activations, and keeps gradients at 32 bits.

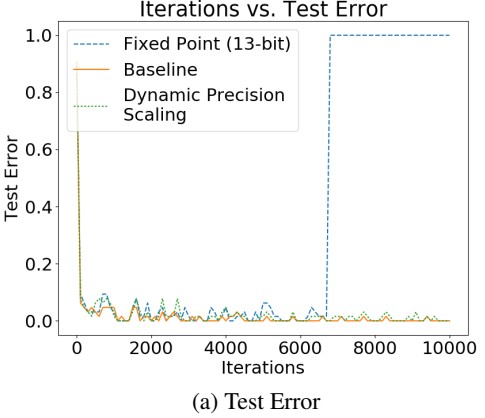
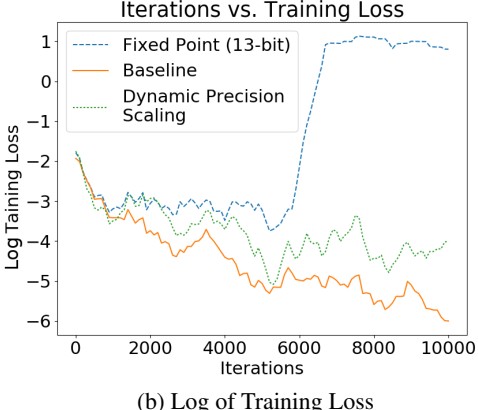

| (a) Test Error | (b) Log of Training Loss |

Figure 1: Comparison of training with Dynamic Precision Scaling vs. the baseline (floating point) vs. fixed point reduced precision (13 bit weights and activations).

Our results reveal that we can achieve accuracy on-par with the baseline, whilst drastically reducing the bit-width used for both weights and activations. Our dynamic precision scaling algorithm in general, however, doesn't reduce the gradient bit-width very much, as this requires the most precision in order for training to converge. The training loss using DPS is, in general, larger than the training loss of the baseline model without hurting accuracy, suggesting that the reduced precision may act as a regularization technique during training – this needs validation via experimentation on larger networks and more complex datasets. Note that naively reducing the bit-width of weights and activations to a fixed 13-bits with no dynamic precision scaling results in the training process failing to converge. With dynamic precision scaling, however, 13-bit weights and activations are sufficient early in the training process.

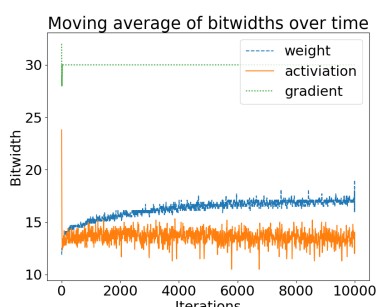

Figure 2: Moving average bitwidths during training using DPS.

## 5 DISCUSSION

We introduce a dynamic precision scaling algorithm that uses quantization error as a metric for scaling dynamic bit-width fixed point values during neural network training. Combining this with stochastic rounding, we achieve greatly reduced bit-width during training, whilst remaining within a fraction of a % of SOTA accuracy on the MNIST dataset. This avenue of algorithmic work, when paired with emerging hardware for training, has the potential to greatly increase the productivity of engineers and machine learning researchers alike by decreasing training time.

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
