# OpenReview forum: "Quantization Error as a Metric for Dynamic Precision Scaling in Neural Net Training"
_ICLR.cc/2018/Workshop — Reject_

### Official Review · AnonReviewer3 · 2018-03-08
**good work with solid experiments**

**Rating:** 7
**Confidence:** 4

**Review:**

The authors proposed a modified fixed point scheme for training neural networks with reduced precision. the integer and fractional part of the fixed point format can be independently adjustable during training.

It is very interesting (and convincing) that the authors showed that 13-bit fixed point explodes at some point of training, while the baseline and the model trained with DPS stay stable throughout the training.

Also, since the bit width for activations/gradients/weights is adaptive during training, the authors showed that gradients always need a high bit width, the activations need a relatively low bit width, while for weights the need for accuracy increases during training.

One caveat is its realization on a real hardware. If the precision of fixed point format always changes, is there an efficient enough way to utilize the benefit of the dynamically reduced bit width? But I agree that this question could be beyond the scope of a workshop paper.

---

### Official Review · AnonReviewer1 · 2018-03-09
**Incremental idea with a weak evaluation**

**Rating:** 5
**Confidence:** 3

**Review:**

This paper proposes an alternative way for dynamic precision scaling of the fixed point representations that depends on quantisation error statistics; the integer and fractional parts are adjusted according to the overflow rate and average quantisation error respectively. This extends the work of Essam et al.  to also include dynamic bit-widths. The authors then employ successfully the proposed scheme for the quantization of  the weights, activations and gradients of a simple LeNet-5 on MNIST.

The paper is reasonable well written and the idea is simple, easy to implement and well framed in the context of prior work. Nevertheless, I do believe that the contribution is incremental and that the experiment section needs more work. For example, comparisons against the relevant literature discussed at Table 1 are missing so it is hard to understand the significance of the contribution. Furthermore, it would be good if the authors could discuss the sensitivity of the method to the E_max and R_max hyper parameters.

Pros:
- Simple idea that seems to work on the toy experiment

Cons:
- Experiment section needs more work to be convincing

---

### Official Review · AnonReviewer2 · 2018-03-09
**This paper proposes a dynamic precision scaling method for training quantized networks. The idea is interesting, however the achieved compression rate is very impressive.**

**Rating:** 4
**Confidence:** 4

**Review:**

The authors propose quantization error as a metric for training networks with dynamic precision. This idea of dynamic precision is interesting, as it may lead to better compression rate. However, the experimental results of this paper is not that promising.

In distribute training, gradient sync requires large network bandwidth. And highly compressed gradient can help a lot. Only compressing weights or activations lightly is not that important, unless we can compress them to much less bits (such as 1-8 bits). Actually, there are already many related works about network quantization (fixed quantization level) that can achieve much better results.

For experiments, only evaluations on MNIST are conducted. The authors are recommended to try larger datasets (eg. imagenet) and more stoa networks (eg. residual nets).

---

### Decision · Program_Chairs · 2018-03-20
**ICLR 2018 Workshop Acceptance Decision**

**Decision:**

Reject

**Comment:**

Based on the reviews, this paper has not been accepted for presentation at the ICLR workshop. However, the conversation and updates can continue to appear here on OpenReview.